# How Glucocorticoids Affect the Neutrophil Life

**DOI:** 10.3390/ijms19124090

**Published:** 2018-12-17

**Authors:** Simona Ronchetti, Erika Ricci, Graziella Migliorati, Marco Gentili, Carlo Riccardi

**Affiliations:** Department of Medicine, Section of Pharmacology, University of Perugia, 06129 Perugia, Italy; erika.ricci@studenti.unipg.it (E.R.); graziella.migliorati@unipg.it (G.M.); marcogentili1988@hotmail.it (M.G.); carlo.riccardi@unipg.it (C.R.)

**Keywords:** glucocorticoids, neutrophils, inflammation, innate immunity

## Abstract

Glucocorticoids are hormones that regulate several functions in living organisms and synthetic glucocorticoids are the most powerful anti-inflammatory pharmacological tool that is currently available. Although glucocorticoids have an immunosuppressive effect on immune cells, they exert multiple and sometimes contradictory effects on neutrophils. From being extremely sensitive to the anti-inflammatory effects of glucocorticoids to resisting glucocorticoid-induced apoptosis, neutrophils are proving to be more complex than they were earlier thought to be. The aim of this review is to explain these complex pathways by which neutrophils respond to endogenous or to exogenous glucocorticoids, both under physiological and pathological conditions.

## 1. Introduction

Cells that are capable of engulfing external pathogens are common among evolutionarily related species. In humans, these cells are called neutrophils, and they represent 50% to 70% of the total leukocytes in blood. The complex role of neutrophils has been discovered over the years, and they have been found to have the dual ability to resolve inflammation and infections, and destroy host tissues. Neutrophils, which are short-lived, adaptable and highly mobile cells of the innate immune system, are released from the bone marrow into the bloodstream after completion of the differentiation process. In the absence of inflammatory signals, they die via apoptosis within 8–12 h. In case of infection or inflammation, their life span is extended up to 1 or 2 days to permit extravasation and migration into inflamed tissues [1,2,3]. As observed in the case of other leukocytes, several neutrophil functions are controlled by glucocorticoids (GCs). These hormones have a wide range of effects in all types of cells and regulate many physiological responses, from glucose metabolism to immune reactions. The modulation of both innate and adaptive immune responses by GCs is necessary for the protection of an organism from harmful external agents and for potentiating innate immune reactions against invading pathogens [4,5,6,7].

In this review, we will focus on the cause-effect relationship between GCs and neutrophils under both normal physiological and pathological conditions, and the pharmacological effect of synthetic GCs on neutrophil responses.

## 2. Role of GCs in the Control of Neutrophil Maturation and Extravasation

Until now, neutrophils were believed to only have a weak response to GCs; however, recent findings have demonstrated that GCs exert fine-tuned control over neutrophil activities. Endogenous GCs are one of the factors that promote the maturation of neutrophils in the bone marrow and favor the mobilization of neutrophils from the bone marrow into circulation [8,9]. This is one of the reasons why the response to excessive GC release, as observed under stress conditions, leads to neutrophilia. However, GCs counteract this effect by tightly regulating neutrophil distribution to prevent inappropriate neutrophil accumulation. The molecular mechanism underlying this control involves a decrease in L-selectin (CD62L) expression on the neutrophil surface, which is most prominent in mature cells [8,10]. L-selectin is one of the three cell adhesion molecules of a family of proteins that includes L-, E- and P-selectin, which are expressed on most circulating leukocytes [11,12]. The shedding of L-selectin leads to faster rolling speeds that favor detachment of the neutrophils from the endothelium into blood flow. This effect can be blocked by inhibitors of p38 Mitogen-activated protein kinase (MAPK), which are important in the intracellular mechanism involved in the control of L-selectin shedding [13]. Another factor involved in this mechanism is Annexin A1 (Anxa1), a well-known GC-induced anti-inflammatory protein, which was demonstrated to reduce the expression of L-selectin on peripheral blood neutrophils, thus favoring neutrophil detachment and mediating the reduction of neutrophil migration induced by GCs [14]. More recently, our group demonstrated that GC induction of the GILZ protein promotes transcription of the Anxa1 gene; this finding implies that Anxa1 is indirectly induced by GCs [15,16]. Furthermore, Anxa1 expression was previously demonstrated to cause the shedding of L-selectin, which leads to reduced or even null accumulation and extravasation of neutrophils into inflamed tissue, which ultimately has an anti-inflammatory effect.

Thus the findings so far indicate that the control of neutrophil function by GCs is complex and starts from their extravasation and distribution.

## 3. Anti-Inflammatory Effects of GCs on Neutrophils

Neutrophil activation is initiated in response to inflammation: this process starts with the recruitment of neutrophils to the inflamed site from blood circulation via their adhesion and migration, under the guidance of chemoattractant gradients [1,17,18]. The mechanisms underlying the process include the binding of P-selectin glycoprotein ligand (PSGL-1) on the neutrophil surface to P- and E-selectins on the endothelial cells, which induces the rolling phase, and the subsequent binding of neutrophil LFA-1 (CD11a/CD18) to its endothelial ligand, Intracellular adhesion molecule-1 (ICAM-1). This process ends with the recognition of pathogen-associated molecular pattern (PAMP) moieties on bacterial cells through pattern recognition receptors (PRRs), such as Toll-like receptors (TLRs). When an inflammation is “sterile”, that is, when it is not caused by pathogens, TLRs can recognize host-derived material from damaged tissues, and initiate the inflammatory response [19]. Once neutrophils enter the interstitial space, they use the gradient of chemoattractants, such as IL-8, to approach invading microbes; upon binding of IL-8 and other chemotactic proteins to specific neutrophil receptors, the signaling cascade of the MAPK/ERK pathway is initiated, leading to an oxidative burst. The parallel stimulation of TLRs leads to the activation of the NADPH oxidase machinery and the production of reactive oxygen species (ROS). All these reactions support the antimicrobial activity of neutrophils, and result in bacterial killing, phagocytosis, degranulation and NETosis (the production of neutrophil extracellular traps). The neutrophils themselves are then eliminated by apoptosis, as described below. The incorrect removal of neutrophils from the inflamed site can hamper the resolution of the inflammatory process, and lead to what is known as a “non-resolving inflammation”, which is very harmful for the host and can lead to chronic inflammation or even cancer [20,21]. In such a context, GCs play a pivotal role in dampening all steps of neutrophil activation: as described earlier, GCs can reduce the expression of L-selectin, on the one hand, and of the adhesion molecules on the endothelium, on the other hand, to prevent neutrophil adhesion and favor their detachment [22,23,24]. GCs can also affect the levels of NADPH oxidase subunits in human neutrophils, since administration of hydrocortisone or stress hormones has been demonstrated to reduce the levels of the p47phox subunit of NADPH oxidase. Further, the synthetic GC dexamethasone has been shown to concomitantly reduce superoxide release and ROS levels [25,26,27]. GCs have also been shown to reduce the expression levels of the inflammatory enzymes COX-2 and iNOS, which are induced by LPS, IL-8, TNFα and IL-1β in neutrophils [4,28,29]. Other specific neutrophil functions, such as chemotaxis and phagocytosis, can also be inhibited by GCs, even though some contradictory results exist in this regard [30,31,32].

A classical and well-established anti-inflammatory effect of GCs is the repression of mRNA transcription of pro-inflammatory genes. This genomic effect has also been observed in neutrophils, of both equine and human origin, in which dexamethasone administration was found to reduce the mRNA levels of IL-1β, TNF-α, and IL-8 and to increase the mRNA levels of glutamine synthetase, an enzyme that plays an essential role in the metabolism of nitrogen [33].

Another anti-inflammatory effect of GCs is their influence on the metabolism of neutrophils via an increase in glucose consumption and decrease in the activity and expression of glycerol-6-phosphate dehydrogenase (G6PDH), which is a well-known substrate for NAPDH-dependent respiratory burst. This effect was demonstrated in a rat model, in which dexamethasone treatment resulted in a decrease in phosphate-dependent glutaminase activity but did not affect the glutamine metabolism in neutrophils [34].

Finally, our group has recently demonstrated that GILZ, a GC-induced protein, restrains neutrophil activation via inhibition of the MAP kinase pathway. This indicates a genomic control of neutrophil activation by GCs and lays the foundation for novel pharmacological approaches to treat neutrophil-derived inflammation [35].

## 4. Pro-Inflammatory Effects of GCs on Neutrophils

In neutrophils, the GC-induced inhibition of transcription factors that control the expression of pro-inflammatory genes is counter-balanced by GC-induced simultaneous inhibition of the factors that control the expression of anti-inflammatory genes. The resulting effect is that neutrophil-driven inflammation cannot be inhibited by GCs to the same extent as it is in other leukocytes [36]. There are several mechanisms underlying this effect. One of the mechanisms is the modulation of the expression levels of the IL-1β receptor (IL-1βR), which was found to be upregulated by GCs in human neutrophils, thus favoring binding with its ligand, the pro-inflammatory cytokine IL-1β [37]. Another mechanism is neutrophil resistance to GC-induced apoptosis, which prolongs their life span (discussed below) [38]. Finally, receptors for leukotrienes are also involved in the pro-inflammatory effect of GCs on neutrophils: for example, BLT1 expression is upregulated by GCs during both maturation and survival, thereby strengthening the functions of neutrophils [39].

A very important GC-induced pro-inflammatory effect that can have serious consequences in pulmonary diseases is the influence on the ability of neutrophils to synthesize sIL-1Ra, which acts as an antagonist against both IL-1β and IL-1R and possesses anti-inflammatory properties. In vitro, it has been shown that Dexamethasone can inhibit both IL-1β and sIL-1Ra, with a marked inhibition of sIL-1Ra, thus altering the pro- and anti-inflammatory balance of these cytokines. Such activity may explain why patients with chronic obstructive pulmonary disease (COPD) who are administered GCs via the nasal route (through inhalation) exhibit low serum levels of sIL-1Ra. However, the role of sIL-1Ra, which is secreted at low levels by neutrophils in the inflammatory environment, still needs to be clarified [40].

## 5. GCs and Neutrophil Apoptosis

It has been long known that GCs prevent apoptosis in neutrophils. Stress conditions or administration of pharmacological doses of GCs cause neutrophilia, which is dependent on both the increased migration of neutrophils from the bone marrow to blood flow and their increased survival [38,41,42]. Several cellular components of intracellular signaling pathways promote the survival of neutrophils, and many of them are targets of the anti-apoptotic effect of GCs. For example, p38 MAP kinase has been demonstrated to participate in the GC-mediated induction of the anti-apoptotic Bcl-2 family member Mcl-1 [43]. The anti-apoptotic role of Mcl-1 has been confirmed by recent studies on human neutrophils, in which GC-induced GILZ expression led to JNK activation and phosphorylation of Mcl-1, which subsequently led to the degradation and consequent apoptosis of neutrophils [44]. The contradictory role of a GC-induced gene in the promotion of neutrophil apoptosis can be explained by a cellular model of GILZ-transfected neutrophil-like cells, which might slightly differ from circulating neutrophils. Alternatively, apoptotic regulation by GCs might be fine-tuned according to the microenvironment: for example, hypoxic conditions may tilt the balance in favor of inflammation and against apoptosis [45].

Regulation of the expression levels of other members of the BcL-2 family by GCs has also been demonstrated: for example, increase and decrease in the expression levels of the pro-survival A1 and the pro-apoptotic Bak, respectively, were observed in DEX-treated bovine neutrophils [46]. In addition, other families of proteins are involved in the control of the apoptosis machinery, including the pro-survival IAP protein family [47]. IAPs are expressed in neutrophils and upregulated during inflammation, and more importantly, their expression is maintained by GCs, thus contributing to survival of neutrophils [43]. Downregulation of pro-apoptotic proteins is another means by which GCs contribute to neutrophil survival. One such important protein in the apoptotic process is Fas, a surface molecule involved in the apoptosis of activated lymphocytes and neutrophils, which was found to be downregulated by GCs in bovine neutrophils [48]. Most of the listed effects are GR-mediated, since the GR inhibitor RU486 has prevented the pro-survival GC effect. However, high levels of expression of the GRβ isoform, with no transcriptional activity, has been speculated to contribute to the anti-apoptotic effects of GCs in neutrophils [49]. Although GCs inhibit spontaneous apoptosis in a concentration-dependent manner, the fine-tuned regulation of apoptosis and survival of neutrophils is not an on-off process, but seems to require a more complex transcriptional regulation mechanism that is dependent on the state of neutrophils (resting/activated) as well as the microenvironment [45,50,51].

## 6. Neutrophils, Disease and GC Treatment

GCs are conventional therapeutic agents for the management of allergic and autoimmune diseases that are associated with persistent inflammation. One such disease is asthma, which is a chronic airway inflammatory disorder for which the mainstay therapy includes inhaled or systemic GCs. Based on the efficacy of the administered steroids, asthmatic patients can be divided into GC-sensitive and GC-resistant patients. In GC-resistant patients, the neutrophilic component is considered to be one of the factors responsible for the failure of the steroid treatment. In fact, increased neutrophil counts have been detected in the sputum, peripheral blood and bronchoalveolar lavage fluid of patients who have severe or steroid-resistant asthma [52,53,54]. Several factors have been found to be associated with neutrophil accumulation: (i) dysbiosis, in which a predominance of specific bacterial species accounts for excessive recruitment of neutrophils [55]; (ii) increased levels of chemokines and pro-inflammatory cytokines, including CXCL1, CXCL5, IL-1β, IFNγ, and IL-6, which promote the recruitment of neutrophils [56]; (iii) high levels of LTB4, a potent chemoattractant and pro-survival factor for neutrophils [57,58]; (iv) inadequate removal of dead neutrophils or inhibition of their apoptosis (although this hypothesis has recently come under question) [54,59]. In this context, GC treatment can exacerbate these events and promote neutrophilia. Excessive neutrophil levels, in addition to causing damage through their microbicidal activity and NET formation, promote airway remodeling, mucus hypersecretion and an overall decline in lung function, thus worsening asthma. However, when neutrophil counts were performed in hospitalized patients who were subjected to GC treatment, the findings showed that neutrophilia was associated with underlying disease rather than GC use [60]. Nonetheless, GCs can influence neutrophil function even in GC-resistant asthmatics, although to a lesser extent than in GC-sensitive patients. For example, the chemotactic IL-8 is similarly inhibited in both types of patients by dexamethasone, but the in vitro addition of asthmatic serum in combination with dexamethasone led to minor IL-8 inhibition in GC-resistant compared to GC-sensitive neutrophils [59]. Furthermore, the level of the phosphatase MKP-1, which dephosphorylates and inactivates MAPKs, was found to have increased to a greater extent in GC-sensitive neutrophils than in GC-resistant neutrophils by dexamethasone treatment, thus accounting for the reduction in pro-inflammatory cytokine expression [59].

Neutrophilic inflammation is a prominent characteristic of another chronic inflammatory disease of the lower airway—COPD. In COPD patients, neutrophils affect the balance between proteases and anti-proteases in favor of lung destruction. In addition, peripheral neutrophils in these patients exhibit the characteristics of activated cells and increase ROS production, which is associated with worsening of the disease [61,62]. Even though therapy with inhaled GCs improves the quality of life of COPD patients, it has little or no effect on the underlying inflammation [63,64]. This resistance seems to depend on multiple factors: one factor is the downregulation of the histone deacetylase HDAC2, which was found to be involved in the GC-mediated anti-inflammatory pathway that controls the transcription of some anti-inflammatory genes, while another factor is reduced GR expression on airway neutrophils, but not on blood neutrophils, which was found to be associated with the poor clinical response to inhaled glucocorticoid therapy in COPD patients [65,66,67]. Moreover, as observed in asthmatic patients, an increase in the GRβ levels was also observed in the neutrophils of COPD patients; this is also believed to contribute to corticosteroid resistance in COPD patients [68].

Rheumatoid arthritis (RA) is another autoimmune disease with neutrophilic involvement, in which neutrophils exhibit an activated phenotype, contributing to the disease pathogenesis by secreting pro-inflammatory cytokines, producing ROS, releasing granules and excessive NET, and destroying the collagen matrix [3,69]. In the pathogenesis of RA, neutrophils exhibit the greatest cytotoxic effect of all participating cells, which is also attributable to their interaction with other cells of the immune system [70]. In the context of RA, it has been extensively reported that treatment with GCs has differential effects on circulating and infiltrating neutrophils. For example, methylprednisolone was previously demonstrated to decrease the ingression of neutrophils into the inflamed joints of RA patients, without affecting neutrophil egress from the joint. Interestingly, it has been found that intraarticular administration of corticosteroids can reduce 5-lipoxygenase-1 expression and thereby decrease their pro-inflammatory products in neutrophils infiltrating the joints [71,72]. Furthermore, intraarticular corticosteroids also lead to reduced expression of synovial S100A12, a pro-inflammatory factor released by activated granulocytes, in the sublining layer [73]. These results argue in favor of the responsiveness of neutrophils to GCs in RA patients, even though GCs are unable to reduce their activation [74]. However, although a substantial proportion of RA patients show a poor response to GC treatment, this response does not seem to be dependent on the neutrophilic component.

While neutrophils play a role in maintaining homeostasis of the gut, they can contribute to the development of tissue injury in the context of increased bacterial invasion, thus worsening the subsequent inflammatory state and initiating colitic pathologies, such as ulcerative colitis. Further, the resulting activated neutrophils are actively recruited from the blood and show prolonged survival or defective apoptosis [75,76]. Therapy with GCs, including the recently reported budesonide that has reduced side effects, can help resolve the underlying inflammatory status [77]. In addition to systemic administration of GCs, local treatment with GCs has also been reported to be successful. It has been demonstrated that prednisolone enema is efficacious in downregulating the factors that favor neutrophil recruitment and their local activation, including the surface activation marker CD66b, thus accounting for the decrease in the number of these cells during remission [76]. However, although GCs are effective, up to 60% of patients are resistant to GC treatment. The resistance depends on several factors—from the enhanced expression of GRβ to pro-inflammatory Th17 cells. However, so far, no study has reported the direct involvement of neutrophils in GC-resistant colitis, although neutrophils expressing GRβ and granulocyte apheresis have proven to be effective for the treatment of Inflammatory Bowel Diseases (IBDs) [78,79,80,81,82].

The efficacy of pharmacological GC therapy is controversial for the treatment of septic shock, a systemic response to a severe infectious disease that can lead to death, in which an increase in the level of cortisol plays a pivotal role in counterbalancing the powerful uncontrolled immune response [83]. Since neutrophils are devoted to pathogen clearance and are the predominant circulating leukocytes in sepsis, they have been studied for their capacity to bind to and respond to GCs in the context of sepsis. The levels of GR have been found to be decreased in neutrophils during sepsis, thus accounting for the reduced response to both endogenous cortisol and exogenous GCs [84,85]. Recently, CD24, a small cell surface protein involved in neutrophil death in a caspase-dependent fashion, was found to be downregulated in neutrophils from septic patients. This decrease in CD24 levels was associated with impaired CD24-mediated cytotoxic response and consequent neutrophil death. In vitro hydrocortisone treatment was able to reduce the CD24 levels, even though other factors also contribute to its downregulation [86]. Therefore, organ failure in critically ill patients due to exacerbation of the inflammatory response is probably the result of the impaired capacity of neutrophils to undergo apoptosis, which along with reduced GR levels, contributes to the unresponsiveness to GCs. Consequently, in the management of sepsis, the resistance and persistence of neutrophils should be carefully considered.

## 7. Conclusions

There is an extensive and growing body of information on the regulation of the complex functions of neutrophils by either endogenous or exogenous GCs. Early and recent studies have reported the sometimes contradictory effects of GCs on neutrophils in terms of the induction or inhibition of apoptosis in these cells. Likewise, other studies have reported that GCs can exert anti-inflammatory or pro-inflammatory effects on neutrophils (Figure 1). These ambiguous effects are probably dependent on the complex features of the inflammatory microenvironment and the diverse context-dependent functions of neutrophils, which can be stimulated by infectious agents or GCs. In addition, the diversity in the neutrophil subtypes can add a specific cell type-dependent response to GCs, which still has not been investigated [87]. Understanding the complex nature of a cell type that has been so far considered simply as a pathogen-killing and suicidal cell will help elucidate the pathogenic mechanisms of neutrophils in several chronic inflammatory and autoimmune diseases and the complex responses of these diseases to steroid hormones and GC treatments.

## Figures and Tables

**Figure 1 ijms-19-04090-f001:**
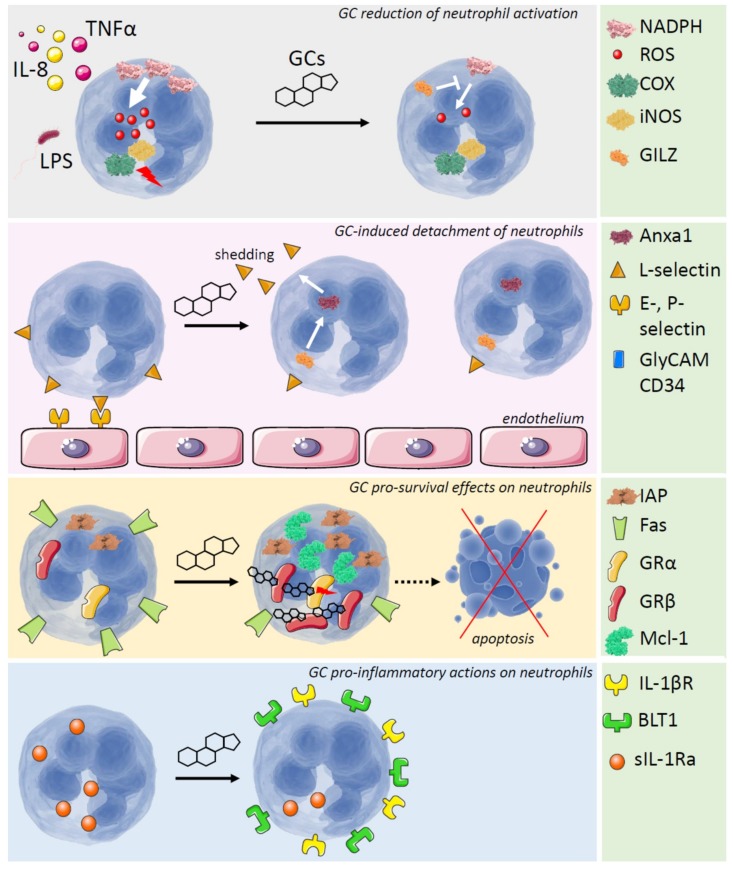
Multiple effects of GCs on neutrophils. From the top to the bottom: (1) the GC-induced anti-inflammatory protein GILZ mediates one of the GC-effects that lead to reduction of neutrophil activation, by inhibiting NADPH-dependent ROS production. COX and iNOS activities are also reduced by GCs; (2) Anxa1, which can be indirectly induced by GCs via GILZ, allows L-selectin shedding, thus favoring detachment of the neutrophils from the endothelium; (3) GCs promote neutrophil survival via several mechanisms: downregulation of the pro-apoptotic surface Fas receptor, upregulation of the pro-survival IAP protein family, upregulation of the anti-apoptotic Mcl-1 protein, and increased levels of the GR-β isoform; (4) the GC pro-inflammatory effects on neutrophils depend on an increase of IL-1βR expression levels, a reduction of sIL-1βRa, and an increase of BLT1 receptor for leukotrienes. White arrows indicate production of ROS or activation of a downstream protein; bars indicate inhibition of a protein function; dotted arrows indicate a prevented effect.

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
