# Peer review of "How Glucocorticoids Affect the Neutrophil Life"

_ijms, 2018, doi:10.3390/ijms19124090_

Reviewer 1 Report

This is an informative and well-written paper that provides a useful insight into the biological life and functionality of neutrophils, and the modifications thereof as a consequence of exposure to glucocorticoids.  There are some minor revisions: define all acronyms at first usage such as G6PDH, ensure correct usage of lower or upper case type, eg p38 MAPK.  For Figure 1, a more detailed summary description under each image would be beneficial to the reader.  Lastly, can the authors comment on the possibility that, given their diverse and sometimes conflicting properties dependent upon the stimulus, there is the possibility of more that one neutrophil population, and if so, is there any evidence of this?

Author Response

There are some minor revisions: define all acronyms at first usage such as G6PDH, ensure correct usage of lower or upper case type, eg p38 MAPK.

We thank the reviewer for his/her valuable suggestions. All necessary acronyms have been defined (page2, lines 48-49) and upper case type checked.

For Figure 1, a more detailed summary description under each image would be beneficial to the reader.

A more detailed description for each image in the figure has been added to the figure legend.

Lastly, can the authors comment on the possibility that, given their diverse and sometimes conflicting properties dependent upon the stimulus, there is the possibility of more that one neutrophil population, and if so, is there any evidence of this?

A brief comment on the existence of neutrophil subpopulations has been added in the Conclusion paragraph, together with a new and updated reference, page 6, lines 258-260.

Reviewer 2 Report

This is an excellent, concise and very well written review.

There are only 2 minor points which might deserve attention:

In line 182, it is stated that "the in vitro addition of asthmatic serum in combination with dexamethasone led to minor IL-8 inhibition in GC-resistant neutrophils". What means minor in this context? Total amount, or compared to GC-sensitive neutrophils?

In the figure, L-selectin and CD62L are represented by different symbols. Are these not only synonyms for the same molecule? If not, what is the difference between CD62L and L-selectin?

Author Response

In line 182, it is stated that "the in vitro addition of asthmatic serum in combination with dexamethasone led to minor IL-8 inhibition in GC-resistant neutrophils". What means minor in this context? Total amount, or compared to GC-sensitive neutrophils?

We thank the reviewer for the useful observations. “Minor” means compared to GC-sensitive neutrophils. This statement has been added in page 4, line 185.

In the figure, L-selectin and CD62L are represented by different symbols. Are these not only synonyms for the same molecule? If not, what is the difference between CD62L and L-selectin?

We thank the reviewer for pinpointing the error about CD62L in the figure. CD62L is indeed L-selectin, thus we removed CD62L with its symbol.